# Evaluation of the Association of Recombinant Proteins NanH and PknG from *Corynebacterium pseudotuberculosis* Using Different Adjuvants as a Recombinant Vaccine in Mice

**DOI:** 10.3390/vaccines11030519

**Published:** 2023-02-23

**Authors:** Nicole Ramos Scholl, Mara Thais de Oliveira Silva, Tallyson Nogueira Barbosa, Rodrigo Barros de Pinho, Mirna Samara Dié Alves, Ricardo Wagner Portela, Vasco Ariston de Carvalho Azevedo, Sibele Borsuk

**Affiliations:** 1Laboratório de Biotecnologia Infecto-Parasitária, Centro de Desenvolvimento Tecnológico, Biotecnologia, UFPel, Pelotas 96010-900, RS, Brazil; 2Laboratório de Imunologia e Biologia Molecular, Instituto de Ciências da Saúde, UFBA, Salvador 40110-100, BA, Brazil; 3Laboratório de Genética Celular e Molecular, Instituto de Ciências Biológicas, UFMG, Belo Horizonte 31270-901, MG, Brazil

**Keywords:** caseous lymphadenitis, aluminum hydroxide, saponin, subunit vaccine, associated antigens, antibodies

## Abstract

Caseous lymphadenitis is a chronic contagious disease that causes economic losses worldwide. Treatments are ineffective, thus demonstrating the importance of vaccination. In this study, rNanH and rPknG proteins from *Corynebacterium pseudotuberculosis* were associated with saponin or aluminum hydroxide adjuvants. Three experimental groups (10 animals each) were immunized with sterile 0.9% saline solution (G1), rNanH + rPknG + Saponin (G2), rNanH + rPknG + Al(OH)_3_ (G3). The mice received two vaccine doses 21 days apart. Animals were challenged 21 days after the last immunization and evaluated for 50 days, with *endpoint* criteria applied when needed. The total IgG production levels of the experimental groups increased significantly on day 42 when compared to the control (*p* < 0.05). When tested against rNanH, G2 had a better rate of anti-rNanH antibodies compared to G3. In the anti-rPknG ELISA, the levels of total IgG, IgG1, and IgG2a antibodies were higher in G2. The vaccines generated partial protection, with 40% of the animals surviving the challenge. The association of recombinant NanH and PknG proteins led to promising protection rates in mice, and although using different adjuvants did not interfere with the survival rate, it influenced the immune response generated by the vaccine formulations.

## 1. Introduction

Caseous lymphadenitis (CLA) is a contagious infectious disease that mainly affects goats and sheep and causes significant economic losses in countries that practice sheep and goat farming [1,2]. CLA is characterized by the appearance of abscesses in the lymph nodes of infected animals, and its etiologic agent, *Corynebacterium pseudotuberculosis*, is a pleomorphic and facultative intracellular bacterium [1,3]. Antibiotic therapy is not an efficient treatment against *C. pseudotuberculosis* because the intracellular form of the pathogen is surrounded by a thick layer of caseous material in infected cells and, therefore, cannot be reached by antibiotics [4]. Thus, prophylaxis is the most efficient control and prevention method, but no commercial vaccine currently provides adequate protection against CLA in sheep and goats [5].

In recent years, different vaccine strategies have been tested against *C. pseudotuberculosis* [6]. For example, the study by Leal et al. (2018) evaluated a vectorized vaccine using recombinant Bacillus Calmette-Guérin (rBCG) expressing *C. pseudotuberculosis* antigens, which induced statistically significant levels of IFN-γ and IL-10, which in turn indicate the formation of a cellular immune response against the pathogen [7]. In the study by Brum et al. (2017), a DNA vaccine that encoded the CP09720 protein sequence was developed; however, the formulation was not effective in inducing an immune response or significant protection after challenge with the virulent strain *C. pseudotuberculosis* MIC-6 [8]. Silva et al. (2018) tested recombinant vaccine formulations in which the rCP01850 and rCP09720 proteins associated with rPLD were used to compose the formulations. Authors reported that the group of animals vaccinated with rPLD + rCP01850 associated with the saponin adjuvant led to high levels of total IgG antibodies, protected 50% of the challenged animals, and induced a Th1-type response characterized by increased production of the cytokines TNF-α and IFN-γ [9].

In these circumstances, several vaccine formulations using protein subunits came to light and were tested for their ability to produce a strong and lasting immune response [6,10]. Genomic studies have identified promising vaccine targets using high-throughput technologies and explored the molecular basis of these targets’ virulence and pathogenicity [11]. In this context, in the last decade, the FRC41 strain of *C. pseudotuberculosis*, which derives from a human isolate, was subjected to a complete analysis of its genome that resulted in the identification of potential virulence factors, for instance, genes coding for NanH (neuraminidase H) and PknG (protein kinase G) proteins [12].

NanH is characterized as a sialidase inserted into the bacterial cell membrane, whose location increases its potential as a vaccine target by facilitating antigen recognition by immune system cells [13]. PknG was characterized in *Mycobacterium tuberculosis* as a protein kinase that acts by preventing phage-lysosomal fusion, allowing the bacteria to survive and multiply within the cell [13]. Thus, the use of recombinant proteins NanH and PknG shows promise. However, to increase the number of antigens available for recognition by the immune system, the use of these proteins in an association is encouraged and is corroborated by previously published studies that used combined antigens in vaccine formulations and reported synergistic and efficient effects [9,14].

In addition, since adjuvants are substances capable of intensifying the immune response if added to vaccine formulations, selecting an adequate adjuvant is as important as selecting immunogenic targets because it can lead to an enhanced protective response, consequently reducing the number of vaccine doses necessary [15,16]. Aluminum hydroxide, an adjuvant widely used in human vaccines, is known for inducing mixed Th1/Th2 immune responses, activating macrophages to participate in the presentation of antigens to immune system cells, and stimulating the production of specific antibodies that are more durable against the antigens present in the vaccine formulations [16]. Saponins, used in human and veterinary medicine, are known to lead to a Th1 response, activate cytotoxic T lymphocytes, and stimulate antibody generation [16,17].

Therefore, in this work, we evaluated whether vaccine formulations composed of the association of recombinant proteins NanH and PknG combined with aluminum hydroxide or saponin adjuvants can protect Balb/c mice against challenge with the virulent MIC-6 strain of *C. pseudotuberculosis.*

## 2. Materials and Methods

### 2.1. Microorganisms

In this study, two *C. pseudotuberculosis* strains were used: the attenuated strain 1002 and the pathogenic strain MIC-6. Both strains were grown in brain-heart infusion (BHI) broth supplemented with 0.5% Tween 80 under agitation or, when needed, in solid BHI (1.5% agar) at 37 °C for 72 h. *Escherichia coli* strains TOP 10 and BL21 (DE3) Star were also used. Both were cultivated under agitation for 16 h at 37 °C in Luria-Bertani (LB) liquid medium and, when necessary, its solid form, with the addition of 1.5% agar for bacteriological use. LB medium may have been supplemented with the antibiotic ampicillin (100 µg/mL).

### 2.2. Cloning, Transformation, and Selection of Recombinant Clones

The previously constructed plasmid pD444-NH/pknG was kindly provided by the Cellular and Molecular Genetics Laboratory of the Federal University of Minas Gerais. The pAE/nanH recombinant plasmid was previously constructed by our research group. Briefly, the nanH gene was amplified by PCR using the following primers: NanHF (5′-ATAGATCTATGCGCTGGTAACACTC-3′) and NanHR (5′-AAGGTACCTTAAAATGCCAGCAG-3′), which contain sites for BglIII and KpnI enzymes. The PCR reaction was performed in a final volume of 50 μL containing 50 ng of genomic DNA from *C. pseudotuberculosis* strain 1002, in addition to 10 µM NanHF and NanHR primers, and Mastermix (Promega). The PCR product and the pAE vector [18] were digested with BglIII and KpnI enzymes and then purified using the GFX PCR DNA and Gel Band Purification Kit (Invitrogen). Ligation of the nanH gene to the pAE vector was performed using 1 U of the enzyme T4 DNA Ligase (Invitrogen). The recombinant plasmid was characterized by cleaving it with the aforementioned restriction enzymes. The recombinant plasmid was named pAE/nanH.

### 2.3. Heterologous Expression of Proteins NanH and PknG from C. pseudotuberculosis in E. coli and Protein Purification

Recombinant plasmids pAE/nanH and pD444-NH/pknG were introduced into *E. coli* BL21 (DE3) Star by heat shock. Protein expression was induced by adding 1 mM of isopropyl α-D-thiogalactoside (IPTG) to cultures that were incubated in an orbital shaker for 3 h at 37 °C. To purify the recombinant proteins, the cultures were centrifuged, and the pellet was resuspended in buffer (50 mM NaH_2_PO_4_; 300 mM NaCl; 20 mM Imidazole; 8 M urea) with the addition of 100 μg/mL of lysozyme, sonicated (cycles), and maintained in refrigeration under stirring at 4 °C for 16 h. Recombinant proteins were purified through affinity chromatography on a HisTrap™ Sepharose nickel column (GE Healthcare) followed by dialysis to allow protein refolding.

### 2.4. Western Blotting

Western blotting was performed to confirm the identity of the recombinant proteins. For this, a 12% SDS-PAGE electrophoresis was performed, and the protein samples in the gel were electro-transferred to a nitrocellulose membrane (Nitrocellulose Blotting Membrane–GE Health Care). A PBS-milk powder solution (5%) was used to block the membrane for 16 h at 4 °C. The membrane was then washed three times with PBS-T (0.05% Tween) under agitation, and the anti-6X His tag monoclonal antibody (Sigma Aldrich) was added at a dilution of 1:4000 in PBS-T. The membrane was kept under agitation for 1 h. Subsequently, the washing step was repeated, and the anti-mouse IgG antibody conjugated with peroxidase (Sigma Aldrich) was added to the membrane at a dilution of 1:4000 in PBS-T, followed by gentle agitation for 1 h. Reactive bands were developed using a solution containing diaminobenzidine (DAB), 0.3% nickel sulfate, 50 mM Tris-HCl buffer (pH 7.6), and H_2_O_2_.

### 2.5. Animals and Ethics Statement

For the immunization and challenge experiment, 30 female Balb/c mice between 6–8 weeks old and susceptible to infection by *C. pseudotuberculosis* were used. The experiment was approved by the Ethics Committee in Animal Experimentation of the Federal University of Pelotas (CEEA/UFPel) under protocol number 12522–2019 and proceeded following the norms of the National Commission for Animal Welfare (COBEA). All animals were maintained under conditions of *ad libitium* food and water supply and adequate states of temperature and humidity in 12-h light and dark cycles.

### 2.6. Immunization and Challenge

The animals were allocated into 3 groups of 10 animals each, as follows: 0.9% Saline Solution (G1), rNanH + rPknG + Saponin (G2), and rNanH + rPknG + Al(OH)_3_ (G3). Group G1 received 0.9% saline solution injections (negative control). Meanwhile, experimental groups G2 and G3 were inoculated with a solution containing 50 μg of each recombinant protein and 7.5 μg of saponin adjuvant as described by Silva et al. (2014) [19] and 15% Al(OH)_3_ as per Brum et al. (2017) [8], respectively. The mice were immunized subcutaneously with two vaccine doses 21 days apart. Blood collections were performed on days 0 (before immunization), 21 (after the first vaccine dose), and 42 (after the second vaccine dose) of the experiment. The animals were challenged 21 days after the last immunization with 2 × 10^4^ CFU/mL (2x the LD_50_) of the virulent *C. pseudotuberculosis* MIC-6 strain injected intraperitoneally and evaluated for 50 days. *Endpoint* criteria were observed [20], and when reached, animals were euthanized by isoflurane anesthesia. In the end, protection was calculated based on animals’ survival rates.

### 2.7. Assessment of the Humoral Immune Response

The humoral immune response was assessed by identifying specific IgG by indirect ELISA. Serum from animals belonging to groups G2 and G3 was submitted for the detection of both antibodies (anti-rNanH and anti-rPknG) separately. The serum from the G1 group was tested against a mix containing the rPknG and rNanH proteins. For this, 96-well polystyrene plates were sensitized with 0.1 μg/well of a recombinant protein (according to the group), diluted in carbonate-bicarbonate buffer (pH 9.8), and incubated for 16 h at 4 °C. Subsequently, the plates were washed three times with PBS-T (1X PBS, pH 7.4, 0.1% Tween 20) and blocked with a 5% solution of milk in PBS-T. After washing the plates again, 100 μL/well of mouse serum (1:50 in PBS-T) was added in duplicate, and the plates were incubated for 1 h. Subsequently, the plates were washed again, and 100 μL/well of anti-mouse IgG antibody conjugated with peroxidase (Sigma Aldrich) was added at a dilution of 1:5000 in PBS-T for the detection of total IgG. For the detection of IgG1 and IgG2a, 100 µL/well of goat anti-mouse IgG1 antibody (diluted 1:5000 in PBS-T) or goat anti-mouse IgG2a antibody (diluted 1:2000 in PBS-T) were added to the plates, and these were incubated for 1 h at 37 °C and subsequently washed. Following this, a peroxidase-conjugated goat anti-IgG antibody (Sigma Aldrich) (1:5000 in PBS-T) was added to the plates (100 μL/well). After incubation (1 h at 37 °C), the plates were washed five times with PBS-T. Then, a developing solution consisting of 0.04% H_2_O_2_ and 0.01 g of o-phenylenediamine dihydrochloride diluted in citrate-phosphate buffer (pH 5.0) was added (100 μL/well), and the plates were incubated at room temperature for 15 min in the dark. Finally, 50 μL/well of stop solution containing 4 N H_2_SO_4_ was added to stop the reaction. Absorbance was measured at 492 nm using a spectrophotometer.

### 2.8. Statistical Analyses

Fisher’s exact test and the log-rank test were used to verify the differences in mortality and survival rates between immunized animals and the control group. Variations in IgG production levels were analyzed using a one-way analysis of variance (ANOVA) followed by Tukey’s post-test. Significant statistical differences were considered at *p* values < 0.05. GraphPad Prism Software version 7 for Windows was used for all statistical analyses.

## 3. Results

### 3.1. Expression, Purification, and Identity Confirmation of Recombinant NanH and PknG Proteins

The proteins were expressed as inclusion bodies in the *E. coli* strain BL21 (DE3) Star and subsequently solubilized in 8 M urea. The expression yields obtained were quantified using a commercial kit based on the bicinchoninic acid kit (BCA; Pierce, USA) and the rNanH and rPknG proteins showed 21.6 mg/L and 6.4 mg/L, respectively. The identity of the recombinant proteins was confirmed by Western blot, in which reactive bands with the expected sizes of approximately 70 kDa for rNanH and 80 kDa for rPknG were observed, as shown in Figure 1 and Appendix A.

### 3.2. Evaluation of the Protective Potential of Vaccine Formulations

Survival rates for mice challenged with the *C. pseudotuberculosis* MIC-6 strain after vaccination schedules described in Topic 2.6 are shown in Figure 2. After the challenge, of the ten animals in G1 (negative control), nine displayed at least one *endpoint* criterion. Both experimental groups, G2 and G3, showed protection levels of 40% but were not statistically different from the control group (*p* = 0.8618).

### 3.3. Assessment of the Humoral Immune Response

The production levels of total IgG, IgG1, and IgG2a against the recombinant proteins NanH and PknG were evaluated through indirect ELISA and are shown in Figure 3 and Figure 4. The results demonstrated that total IgG production rates increased significantly (*p* < 0.0001) in both experimental groups on day 42 when compared to the negative control. When tested against rNanH, G3 showed an increase in total IgG production that started on day 21 and reached its peak on day 42, while G2 showed an increase in production only on day 42 (Figure 3A). As for isotype production rates, G2 obtained better rates for both IgG1 and IgG2a anti-rNanH antibodies (*p* < 0.0001) when compared to G3 (Figure 3B,C), and IgG2a was the most produced isotype in both experimental groups (*p* < 0.001 for G2 and *p* = 0.0071 for G3) (Figure 3D).

In the anti-rPknG ELISA, the levels of total IgG (Figure 4A), IgG1 (*p* = 0.0264) (Figure 4B), and IgG2a (*p* < 0.001) (Figure 4C) antibodies were higher in G2 than G3, and in G2 the IgG2a isotype showed a higher production level compared to IgG1 (*p* < 0.0001) (Figure 4D). Finally, the results of the anti-rNanH and anti-rPknG ELISA assays showed that on day 42, both experimental groups (G2 and G3) had significantly higher antibody levels (*p* < 0.05) than the control group (G1) (Figure 3A–C and Figure 4A–C).

## 4. Discussion

In the present work, we assessed the immunogenic potential and protective efficacy of the recombinant proteins NanH and PknG from *C. pseudotuberculosis*. These recombinant proteins were previously evaluated as possible antigens through bioinformatics analysis and were considered good targets for immunoprophylaxis against the infection caused by *C. pseudotuberculosis* in a murine model [12,13], encouraging their use in vaccine formulations against CLA.

Our results showed that both the group inoculated with rNanH + rPknG + Saponin (G2) and with rNanH + rPknG + Al(OH)3 (G3) protected 40% of the animals challenged with *C. pseudotuberculosis* virulent strain, but were not significantly different from the control group (G1). This can be explained by the fact that in G1, inoculated with 0.9% saline solution, one animal did not meet the *endpoint* criteria, remaining alive until the end of the experiment. For this reason, the vaccine formulations in this study could not generate statistically significant protection in animals challenged with the virulent MIC-6 strain of *C. pseudotuberculosis*. Nonetheless, a 4 in 10 animal survival rate should still be considered a promising potential for a vaccine target. In the study by Droppa-Almeida et al. (2016), the authors tested a recombinant subunit vaccine based on the rCP40 protein against CLA, and the control group also had animals that survived [21], leading to results that were similar to the ones found here and corroborating our hypothesis for these findings. Moreover, the recombinant proteins NanH and PknG have been individually combined with the saponin adjuvant in formulations that obtained significant protection results in up to 60% of the animals [22], demonstrating that these proteins have protective potential and should be further explored as vaccine targets.

In this work, the rNanH and rPknG proteins were used in combination in vaccine formulations. Silva et al. (2018) reported the advantage of using associated proteins by showing that the combination of rPLD and rCP01850 proteins from *C. pseudotuberculosis* and the saponin adjuvant led to the protection of 50% of vaccinated animals [9]. The benefits of using vaccine formulations containing more than one antigen were also demonstrated in the study by Moreira et al. (2022). The authors co-administered the recombinant proteins SpaC, NanH, SodC, and PLD from *C. pseudotuberculosis* in association with the inactivated T1 strain of *C. pseudotuberculosis* to immunize sheep and obtained high levels of IgG production, which remained stable even after the animals were challenged. They also showed that administering a booster dose containing the NanH protein induced an increase in antibody production [14]. Although our study was performed in a murine model, our results did not show such a high level of protection after immunization. We believe that the cytoplasmic location of rPknG may have reduced its processing and presentation to MHC class I, thus hindering the development of an immune response and consequently the protection of challenged animals [13]. Added to that, we have witnessed survival in the control group, which directly interfered with our results for the protection of challenged animals. Nonetheless, this study and previous ones [10,15,22] show that recombinant *C. pseudotuberculosis* proteins are promising vaccine targets that need to be further evaluated, especially if a mixture of these proteins can stimulate different immune responses, a hypothesis that should be further explored in future studies. Therefore, the use of these proteins in vaccine formulations should not be revoked.

In addition to the association of antigens, this study also focused on the use of different adjuvants to compose vaccine formulations. The adjuvants chosen for this study were aluminum hydroxide and saponin. In the study by Rezende et al. (2020), the adjuvant aluminum hydroxide, when associated with rCP01850 from *C. pseudotuberculosis*, was shown to generate a mixed immune response of the Th1 and Th2 types [23]. Bastos et al. (2012) had previously reported that formulations that generate a mixed response are interesting because, although the cellular immune response is the main one against *C. pseudotuberculosis*, developing a humoral response helps in the defense against the pathogen through the neutralizing antibodies [17]. Furthermore, several studies have already used the saponin adjuvant associated with *C. pseudotuberculosis* antigens. The study by Droppa-Almeida et al. (2021) showed that the saponin adjuvant was associated with peptides from the CP40 protein of *C. pseudotuberculosis* [24]. Moreover, in the studies by Droppa-Almeida et al. (2016) and Silva et al. (2014), vaccines for CLA containing recombinant antigens associated with saponin protected 90% and 100% of immunized animals, respectively [19,21]. Sun (2009) suggested that, considering that saponins can stimulate a Th1 response and induce cytotoxic lymphocytes, their use as an adjuvant is ideal for subunit vaccines against intracellular pathogens [25].

In our study, the experimental groups G2 and G3 exhibited a significant production of total IgG and its isotypes IgG1 and IgG2a, with the latter having the highest production levels when tested against both proteins. These results point to the development of a mixed immune response with a tendency toward the Th1 type [26]. Cells arising from the Th1-type response are the main ones involved in immunity against intracellular pathogens [27]. For instance, Th1 lymphocytes produce IFN-γ which modulates antibody class-switch recombination to the IgG2a type and also induces macrophage activation, generating increased phagocytosis [28]. Previous studies have already shown that IgG2a levels associated with the Th1 response highly correlate with phagocytic capacity [26]. Several studies using recombinant *C. pseudotuberculosis* proteins to compose vaccine formulations also reported significant levels of IgG2a production leading to the Th1-type immune response and generating protection in challenged animals [9,24,29], corroborating the results found here. Finally, even though the G2 and G3 groups had the same number of animals protected against the challenge when comparing IgG levels, the G2 group elicited the highest levels of antibodies, and that can be attributed to the use of the saponin adjuvant.

## 5. Conclusions

Our data show the vaccine formulations were able to produce partial protection in challenged animals while generating significant levels of total IgG, IgG1, and IgG2a antibodies against *C. pseudotuberculosis*. The antibodies produced characterized a mixed humoral response with a tendency toward the Th1-type due to the higher levels of the IgG2a isotype. We further characterized the rNanH and rPknG proteins as promising vaccine targets, taking one step forward in vaccine development against CLA. Evaluating the cellular immune response triggered by these formulations is the next step in elucidating whether the association of recombinant proteins NanH and PknG is the right path in vaccine development for CLA, especially if the proteins manage to elicit different immune responses. Finally, testing the association of rNanH and rPknG in combination with other adjuvants is also an important step toward finding the best vaccine formulation against CLA.

## Figures and Tables

**Figure 1 vaccines-11-00519-f001:**
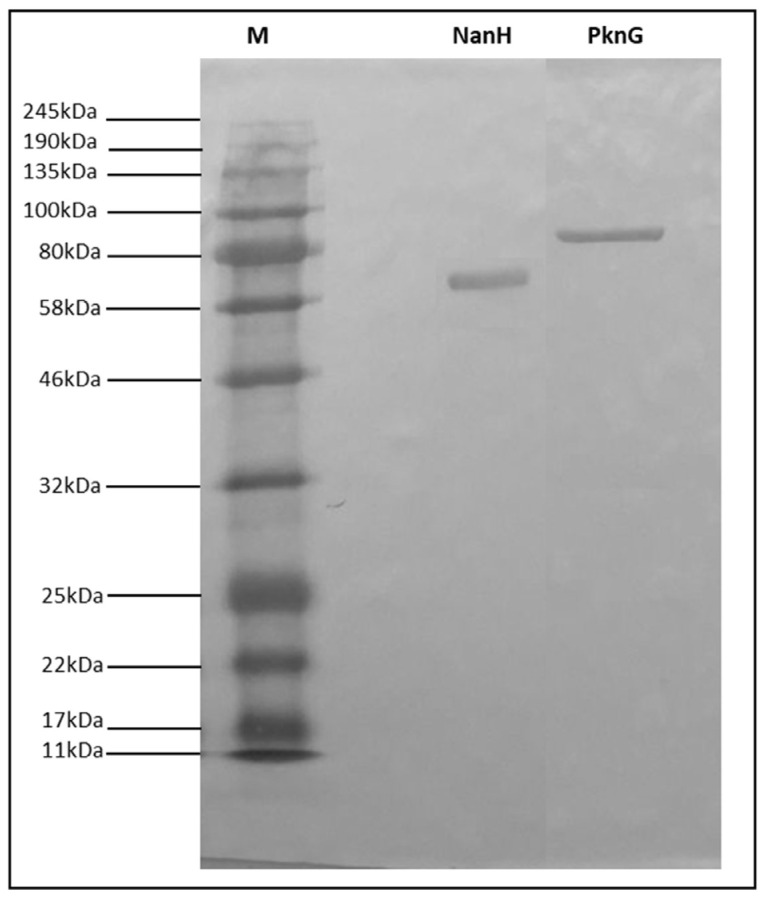
Identity characterization of the recombinant proteins NanH and PknG. Western Blot analysis of NanH and PknG proteins using an anti-6x His tag monoclonal antibody (Sigma Aldrich). (1) Pre-stained protein ladder; (2) purified rNanH; and (3) purified rPknG. rNanH and rPknG are shown as reactive bands of approximately 70 kDa (2) and 80 kDa (3), respectively.

**Figure 2 vaccines-11-00519-f002:**
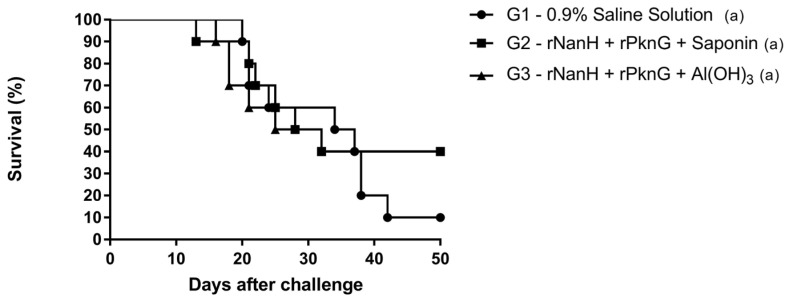
Survival rates of Balb/c mice immunized with rNanH + rPknG proteins and different adjuvants after challenge with the virulent *C. pseudotuberculosis* MIC-6 strain. Data were obtained using ten mice for each group, which were monitored for 50 days after the challenge. Survival curves were compared using log-rank analysis and Fisher’s exact test and considered statistically significant when *p* < 0.05.

**Figure 3 vaccines-11-00519-f003:**
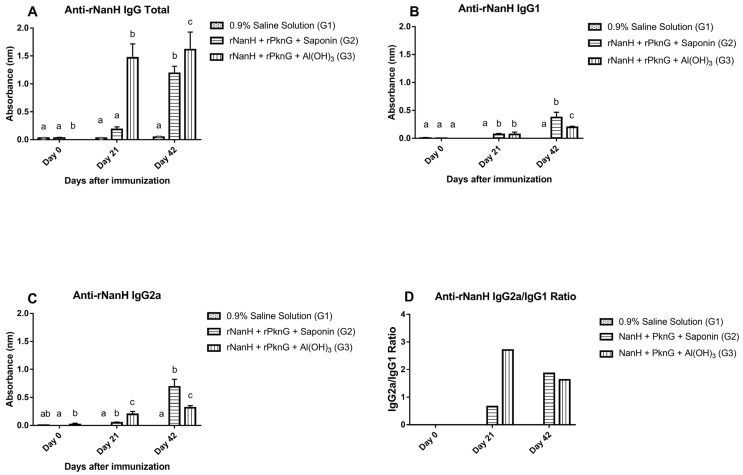
Determination of total IgG, IgG1, and IgG2a levels generated against the *C. pseudotuberculosis* rNanH protein on days 0 (preimmune), 21 (first immunization), and 42 (second immunization). Results are given as mean (bars) ± standard deviation of absorbance (nm) and each experimental group contained 10 animals. Different letters on the same day represent groups with statistically significant differences (*p* < 0.05) while the same letter shows no statistical difference (*p* > 0.05). All groups were compared with each other on a given day. Data from different days were not compared with each other. (**A**) Total anti-rNanH IgG; (**B**) anti-rNanH IgG1; (**C**) anti-rNanH IgG2a; (**D**) IgG2a/IgG1 anti-rNanH ratio.

**Figure 4 vaccines-11-00519-f004:**
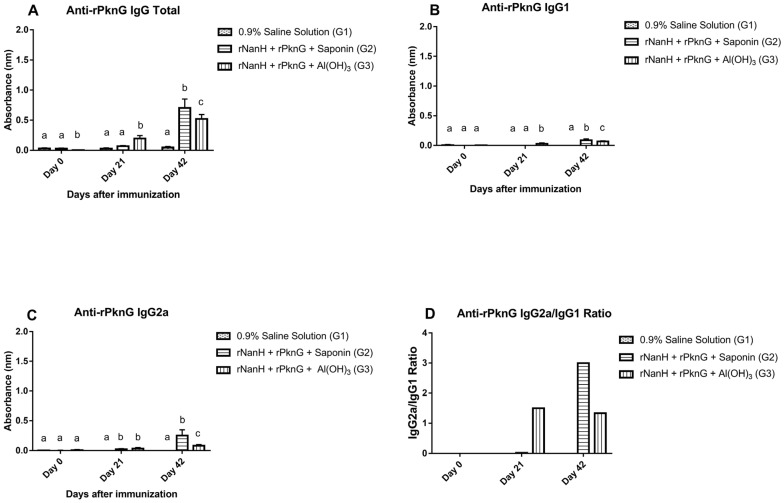
Determination of total IgG, IgG1, and IgG2a levels generated against the *C. pseudotuberculosis* rPknG protein on days 0 (preimmune), 21 (first immunization), and 42 (second immunization). Results are given as mean (bars) ± standard deviation of absorbance (nm) and each experimental group contained 10 animals. Different letters on the same day represent groups with statistically significant differences (*p* < 0.05) while the same letter shows no statistical difference (*p* > 0.05). All groups were compared with each other on a given day. Data from different days were not compared with each other. (**A**) Total anti-rPkng IgG; (**B**) anti-rPkng IgG1; (**C**) anti-rPkng IgG2a; (**D**) IgG2a/IgG1 anti-rPkng ratio.

## Data Availability

Not applicable.

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
