# Peer review of "Evaluation of the Association of Recombinant Proteins NanH and PknG from *Corynebacterium pseudotuberculosis* Using Different Adjuvants as a Recombinant Vaccine in Mice"

_vaccines, 2023, doi:10.3390/vaccines11030519_

Round 1

Reviewer 1 Report

The manuscript “Evaluation of the association of recombinant proteins NanH and PknG from Corynebacterium pseudotuberculosis using different adjuvants as a recombinant vaccine in mice” assessed whether vaccine formulations composed of the association of recombinant proteins NanH and PknG combined with adjuvants can protect animals from C. pseudotuberculosis. The study is important as the first study to immunogenic potential and protective efficacy of recombinant C. pseudotuberculosis proteins.

The work is interesting and precedent, although it is preliminary.

I think it is acceptable after some revision, taking into account the following points.

Minor points:

1.    Section 2.4, Western blotting was used to confirm recombinant C. pseudotuberculosis proteins. It would be nice to show the acquired images, included in the SI.

2.    Figure 2. The current y axis unit is absorbance (nm). With standard curve, the author should be able to convert absorbance to concentration (for example ng/mL), which would allow the audience to better appreciate their results.

Author Response

Reviewer 1
English language and style: (x) English language and style are fine/minor spell check required

Answer: We thank the reviewer for the comment. We have performed a final English review of the manuscript to ensure better readability and grammar accuracy. However, if you still feel like another review is in order, please let us know and we will be happy to do so.

Are the results clearly presented? (x) Can be improved
Answer: The authors have made some changes in how the results are described and added a figure to illustrate one of the results mentioned (Figure 1, page 5, line 198) to improve the results section. We hope the changes were enough to achieve that goal, but if not, we are more than willing to make further improvements. Just let us know, please.

Comments and Suggestions for Authors
The manuscript “Evaluation of the association of recombinant proteins NanH and PknG from Corynebacterium pseudotuberculosis using different adjuvants as a recombinant vaccine in mice” assessed whether vaccine formulations composed of the association of recombinant proteins NanH and PknG combined with adjuvants can protect animals from C. pseudotuberculosis. The study is important as the first study to immunogenic potential and protective efficacy of recombinant C. pseudotuberculosis proteins.

The work is interesting and precedent, although it is preliminary. I think it is acceptable after some revision, taking into account the following points.

Answer: We would first like to thank you for your comments about our work. We have done our best to make the manuscript clearer and more objective and we hope that was what you were hoping for. Before answering your comments one by one, we would like to give a brief description of the changes made in the text. The authors added a Western blotting figure to better illustrate the results (Figure 1, page 5, line 198) and consequently, all the other figures had their numbers changed as follows: Previous Figure 1 → Figure 2 (Page 5, line 211); Previous Figure 2 → Figure 3 (Page 6, line 235); Previous Figure 3 → Figure 4 (Page 7, line 245). The authors also changed sentences as requested by the editorial office to improve readability. These changes are highlighted in green in the “tracked changes” file and can be found as follows:
Materials and Methods section:
1. 2.1 Microorganisms (Page 2, lines 91-95).
2. 2.3 Heterologous expression of proteins NanH and PknG from C. pseudotuberculosis in E. coli and protein purification (Page 3, lines 115-116)
3. 2.6. Immunization and challenge (Page 4, lines 150-151)
4. 2.7 Assessment of the humoral immune response (Page 4, lines 169-171 and 176-183)
5. 2.8 Statistical analyses (Page 4, lines 185-187)
Results section:

6. 3.3 Assessment of the humoral immune response (Pages 6 and 7, lines 239-240 and 249-250)

1. Section 2.4, Western blotting was used to confirm recombinant C. pseudotuberculosis proteins. It would be nice to show the acquired images, included in the SI.

Answer: Thank you for the suggestion. We decided to add the Western blotting figure in the manuscript as mentioned above and we did so by adding it to the main text. The figure was added in topic 3.1. “Expression, purification, and identity confirmation of recombinant NanH and PknG proteins’’ (line 198) and captioned “Figure 1: Identity characterization of the recombinant proteins NanH and PknG. Western Blot analysis of NanH and PknG proteins using an anti-6x His tag monoclonal antibody (Sigma Aldrich). (1) Pre-stained protein ladder; (2) purified rNanH; and (3) purified rPknG. rNanH and rPknG are shown as reactive bands of approximately 70 kDa (2) and 80 kDa (3), respectively.” (Page 5, lines 201-204). The authors hope you approve of the decision to leave it on the main text. If not, please let us know.

2. Figure 2. The current y axis unit is absorbance (nm). With standard curve, the author should be able to convert absorbance to concentration (for example ng/mL), which would allow the audience to better appreciate their results.

Answer: We thank you for your suggestions and we would like to explain why we chose absorbance (nm) as the unit for the y-axis. We took previously published articles as our guide when generating the images. The articles published between 2014 and 2022 (references below) that applied the ELISA methodology to identify antibodies against C. pseudotuberculosis used the absorbance of the samples in nm as the y-axis. For instance, the article by Moreira et al. (2022), published at Vaccines (MDPI) and used as a reference in our manuscript, used absorbance as their y-axis. Moreover, our research group published similar works between 2017 and 2021 using absorbance as the unit as well. Therefore, we would like to keep the absorbance as the unit for now to maintain the standard found in the articles published so far. We hope the reviewer can understand and accept our request. If that is non-negotiable, please let us know and we will make the appropriate changes.

Silva, J.W.; Droppa-Almeida, D.; Borsuk, S.; Azevedo, V.; Portela, R.W.; Miyoshi, A.; Rocha, F.S.; Dorella, F.A.; Vivas, W.L.; Padilha, F.F.; et al. Corynebacterium Pseudotuberculosis Cp09 Mutant and Cp40 Recombinant Protein Partially Protect Mice against Caseous Lymphadenitis. BMC Vet. Res. 2014, 10, 1–8, doi:10.1186/s12917-014-0304-6.

Droppa-Almeida, D.; Vivas, W.L.P.; Silva, K.K.O.; Rezende, A.F.S.; Simionatto, S.; Meyer, R.; Lima-Verde, I.B.; Delagostin, O.; Borsuk, S.; Padilha, F.F. Recombinant CP40 from Corynebacterium Pseudotuberculosis Confers Protection in Mice after Challenge with a Virulent Strain. Vaccine 2016, 34, 1091–1096, doi:10.1016/j.vaccine.2015.12.064.

Brum, A.A.; Rezende, A. de F.S.; Brilhante, F.S.; Collares, T.; Begnine, K.; Seixas, F.K.; Collares, T.V.; Dellagostin, O.A.; Azevedo, V.; Santos, A.; et al. Recombinant Esterase from Corynebacterium Pseudotuberculosis in DNA and Subunit Recombinant Vaccines Partially Protects Mice against Challenge. J. Med. Microbiol. 2017, 66, 635–642, doi:10.1099/jmm.0.000477.

Silva, M.T. de O.; Bezerra, F.S.B.; de Pinho, R.B.; Begnini, K.R.; Seixas, F.K.; Collares, T.; Portela, R.D.; Azevedo, V.; Dellagostin, O.; Borsuk, S. Association of Corynebacterium Pseudotuberculosis Recombinant Proteins RCP09720 or RCP01850 with RPLD as Immunogens in Caseous Lymphadenitis Immunoprophylaxis. Vaccine 2018, 36, 74–83, doi:10.1016/j.vaccine.2017.11.029.

Rezende, A.F.S.; Brum, A.A.; Bezerra, F.S.B.; Braite, D.C.; Sá, G.L.; Thurow, H.S.; Seixas, F.K.; Azevedo, V.A.C.; Portela, R.W.; Borsuk, S. Assessment of the Acid Phosphatase CP01850 from Corynebacterium Pseudotuberculosis in DNA and Subunit Vaccine Formulations against Caseous Lymphadenitis. Arq. Bras. Med. Vet. e Zootec. 2020, 72, 199–207, doi:10.1590/1678-4162-10790.

Droppa-Almeida, D.; Da Silva, G.A.; Do Amorim Costa Gaspar, L.M.; Pereyra, B.B.S.; Nascimento, R.J.M.; Borsuk, S.; Franceschi, E.; Padilha, F.F. Peptide Vaccines Designed with the Aid of Immunoinformatic against Caseous Lymphadenitis Promotes Humoral and Cellular Response Induction in Mice. PLoS One 2021, 16, 1–18, doi:10.1371/journal.pone.0256864.

Barral, T.D.; Kalil, M.A.; Mariutti, R.B.; Arni, R.K.; Gismene, C.; Sousa, F.S.; Collares, T.; Seixas, F.K.; Borsuk, S.; Estrela-Lima, A.; et al. Immunoprophylactic Properties of the Corynebacterium Pseudotuberculosis-Derived MBP:PLD:CP40 Fusion Protein. Appl. Microbiol. Biotechnol. 2022, 8035–8051, doi:10.1007/s00253-022-12279-1.

Moreira, L.S.; Lopes, N. da R.; Pereira, V.C.; Andrade, C.L.B.; Torres, A.J.L.; Ribeiro, M.B.; Freire, S.M.; Santos, R.M. dos; D’ávila, M.; Nascimento, R.M.; et al. The Association of Bacterin and Recombinant Proteins Induces a Humoral Response in Sheep against Caseous Lymphadenitis. Vaccines 2022, 10, doi:10.3390/vaccines10091406.

Reviewer 2 Report

The submitted manuscript by Borsuk et al, provide valuable evaluation of recombinant vaccine of associated rNanH and rPknG antigens from Corynebacterium pseudotuberculosis with different adjuvant within mice model. Achieving animal protection against the challenge and triggered humoral immune responses have presented such approach promising for future development. The manuscript is well-written and controversial data are well-explained and rationalized. Minor comments are suggested prior publication:

1.       Authors should provide references for the adopted concentration of the adjuvants being used in the study.

2.       Images for the performed Western blot membrane should be provided within the Supplementary materials.

3.       Figures 2 and 3, authors should specify whether different letters on same day represent significant differences for each group against the negative control (G1) only or there are another annotations for differences between the treated ones on the same day and different days.

4.       Levels of statistical significance (the obtained P-value) for each comparative group should be provided within the context.

5.       Authors examined associated proteins within immunization and potential antagonism was suggested when findings were compared to reported data of individual protein. However, ensuring this experimentally by the authors would provide more solid evidence, particularly as the authors experienced survival within the control group.

6.       Additionally, the use of associated proteins could be suggested  beneficial particularly if each could mount different immune responses. Authors should emphasize on this while planning for future study investigating the cellular immune responses.

7.       Minor typo should be corrected; like C. Pseudotuberculosis should be italic at lines 45, 47, and 52.

Author Response

A point-by-point response to reviewers ID manuscript vaccines-2210461: “Evaluation of the association of recombinant proteins NanH and PknG from Corynebacterium pseudotuberculosis using different adjuvants as a recombinant vaccine in mice”

Nicole Ramos Scholl 1, Mara Thais de Oliveira Silva 1, Tallyson Barbosa Nogueira 1, Rodrigo Barros de Pinho 1, Mirna Samara Dié Alves 1, Ricardo Wagner Portela 2, Vasco Ariston de Carvalho Azevedo 3, Sibele Borsuk1*

Reviewer 2
English language and style: (x) English language and style are fine/minor spell check required
Answer: We thank the reviewer for the comment. We have performed a final English review of the manuscript to ensure better readability and grammar accuracy. However, if you still feel like another review is in order, please let us know and we will be happy to do so.

Are the methods adequately described? (x) Can be improved
Answer: The authors made punctual changes in the Materials and Methods section for a better understanding of the text without changing what we wanted to say. We hope it is what was expected, but if further changes are needed please let us know and we'll be happy to make them.

Are the results clearly presented? (x) Can be improved
Answer: The authors have made some changes in how the results are described and added a figure to illustrate one of the results mentioned (Figure 1, page 5, line 198) to improve the results section. We hope the changes were enough to achieve that goal, but if not, we are more than willing to make further improvements. Just let us know, please.

The submitted manuscript by Borsuk et al, provide valuable evaluation of recombinant vaccine of associated rNanH and rPknG antigens from Corynebacterium pseudotuberculosis with different adjuvant within mice model. Achieving animal protection against the challenge and triggered humoral immune responses have presented such approach promising for future development. The manuscript is well-written and controversial data are well-explained and rationalized. Minor comments are suggested prior publication:

Answer: We would first like to thank you for your comments about our work. We have done our best to make the manuscript clearer and more objective and we hope that was what you were hoping for. Before answering your comments one by one, we would like to give a brief description of the changes made in the text. The authors added a Western blotting figure to better illustrate the results (Figure 1, page 5, line 198) and consequently, all the other figures had their numbers changed as follows: Previous Figure 1 → Figure 2 (Page 5, line 211); Previous Figure 2 → Figure 3 (Page 6, line 235); Previous Figure 3 → Figure 4 (Page 7, line 245). The authors also changed sentences as requested by the editorial office to improve readability. These changes are highlighted in green in the “tracked changes” file and can be found as follows:

Materials and Methods section:
1. 2.1 Microorganisms (Page 2, lines 91-95).
2. 2.3 Heterologous expression of proteins NanH and PknG from C. pseudotuberculosis in E. coli and protein purification (Page 3, lines 115-116)
3. 2.6. Immunization and challenge (Page 4, lines 150-151)
4. 2.7 Assessment of the humoral immune response (Page 4, lines 169-171 and 176-183)
5. 2.8 Statistical analyses (Page 4, lines 185-187)
Results section:

6. 3.3 Assessment of the humoral immune response (Pages 6 and 7, lines 239-240 and 249-250)

1. Authors should provide references for the adopted concentration of the adjuvants being used in the study.

Answer: We would like to thank you for the suggestion. We have added to the text the references relating to the concentrations of adjuvants used in the vaccine formulations as suggested. The authors would like to clarify that the saponin concentration (7.5 μg) was used according to a work published by Silva et al. (2014). Meanwhile, the aluminum hydroxide concentration (15 %) was based on the work by Brum et al. (2017), one of your research group’s publications. In the text, the changes are as follows: “… 7.5 μg of saponin adjuvant as described by Silva et al. (2014) [19] and 15 % Al(OH)3 as per Brum et al. (2017) [8], respectively” and can be found in page 4, lines 152 and 153. Silva, J.W.; Droppa-Almeida, D.; Borsuk, S.; Azevedo, V.; Portela, R.W.; Miyoshi, A.; Rocha, F.S.; Dorella, F.A.; Vivas, W.L.; Padilha, F.F.; et al. Corynebacterium Pseudotuberculosis Cp09 Mutant and Cp40 Recombinant Protein Partially Protect Mice against Caseous Lymphadenitis. BMC Vet. Res. 2014, 10, 1–8, doi:10.1186/s12917-014-0304-6. Brum, A.A.; Rezende, A. de F.S.; Brilhante, F.S.; Collares, T.; Begnine, K.; Seixas, F.K.; Collares, T.V.; Dellagostin, O.A.; Azevedo, V.; Santos, A.; et al. Recombinant Esterase from Corynebacterium Pseudotuberculosis in DNA and Subunit Recombinant Vaccines Partially Protects Mice against Challenge. J. Med. Microbiol. 2017, 66, 635–642, doi:10.1099/jmm.0.000477.

2. Images for the performed Western blot membrane should be provided within the Supplementary materials.

Answer: Thank you for the suggestion. We decided to add the Western blotting figure in the manuscript as mentioned above and we did so by adding it to the main text. The figure was added in topic 3.1. “Expression, purification, and identity confirmation of recombinant NanH and PknG proteins’’ (line 198) and captioned “Figure 1: Identity characterization of the recombinant proteins NanH and PknG. Western Blot analysis of NanH and PknG proteins using an anti-6x His tag monoclonal antibody (Sigma Aldrich). (1) Pre-stained protein ladder; (2) purified rNanH; and (3) purified rPknG. rNanH and rPknG are shown as reactive bands of approximately 70 kDa (2) and 80 kDa (3), respectively.” (Page 5, lines 201-204). The authors hope you approve of the decision to leave it on the main text. If not, please let us know.

3. Figures 2 and 3, authors should specify whether different letters on same day represent significant differences for each group against the negative control (G1) only or there are another annotations for differences between the treated ones on the same day and different days.

Answer: We thank the reviewer for the suggestion. To clarify, multiple comparisons for a given day (Tukey’s post-test) were made between all groups and not only between each group versus the control. Therefore, if on a given day you have three different letters, all groups are different from each other while those groups which share the same letters are not significantly different. Besides that, no comparisons were made between the data obtained from different days. We have changed the figures’ captions to read as follows: “Different letters on the same day represent groups with statistically significant differences (p < 0.05) while the same letter shows no statistical difference (p > 0.05). All groups were compared with each other on a given day. Data from different days were not compared with each other” and can be found in lines 240-243 (Figure 3) and lines 250-253 (Figure 4). We hope this change has clarified how the analyses were made and improved the general understanding of the figures and results. If not, please let us know and we will make further modifications.

4. Levels of statistical significance (the obtained P-value) for each comparative group should be provided within the context.
Answer: Thank you for the suggestion. We have added the p-values throughout the text where we deemed necessary. More specifically, in topic 3.2 (Evaluation of the protective potential of vaccine formulations) the p-value was added in the sentence “Both experimental groups, G2 and G3, showed protection levels of 40 % but were not statistically different from the control group (p = 0.8618)” in page 5, lines 208-210. Also, the caption for Figure 2 now reads: “Survival curves were compared using log-rank analysis and Fisher's exact test and considered statistically significant when p < 0.05” on page 5, lines 215-217.
Moreover, in topic 3.3 (Assessment of the humoral immune response) the p-value was added to the sentence “The results demonstrate that total IgG production rates increased significantly (p<0.0001) in both experimental groups on day 42” page 6, lines 221-222 and we also added the sentence “when compared to the negative control” in line 222. Finally, p-values were also added in lines 226, 227, 228, 229, 230, and 231 and the captions for Figures 3 (line 241) and 4 (line 251).

5. Authors examined associated proteins within immunization and potential antagonism was suggested when findings were compared to reported data of individual protein. However, ensuring this experimentally by the authors would provide more solid evidence, particularly as the authors experienced survival within the control group.

Answer: We kindly thank you for your comment and suggestion. We review the sentence mentioned in your comment and since we currently would not be able to perform the experiments needed to provide solid evidence for our hypothesis, we decided to modify the sentence to read as follows: “We believe that the cytoplasmic location of rPknG may have reduced its processing and presentation to MHC class I, thus hindering the development of an immune response and consequently the protection of challenged animals [13], and added to that, we have witnessed survival in the control group, which directly interfered in our results for protection of challenged animals” on page 8, lines 289-294. Just as a reminder, the former sentence read as follows: “Although our study was performed in a murine model, our results did not show such a level of protection after immunization, and the data led us to believe that this may be partially due to antagonistic interactions between the proteins because when tested individually by Silva et al. (2020), rNanH and rPknG induced higher protection rates in challenged animals [21]. Moreover, the cytoplasmic location of rPknG may have reduced its processing and presentation to MHC class I, thus hindering the development of an immune response and consequently the protection of challenged animals [13]”.

6. Additionally, the use of associated proteins could be suggested beneficial particularly if each could mount different immune responses. Authors should emphasize on this while planning for future study investigating the cellular immune responses.

Answer: We would like to thank you for the suggestion. We will certainly keep that in mind for the next set of experiments with these (and other) proteins. Also, we decided to make a few changes to the text, more specifically to the sentences that mention the benefits of protein association and our plans for performing cellular immune response assays. In topic 4 (Discussion), lines 294-298 now read: “Nonetheless, this study and previous ones [10,15,22] show that recombinant C. pseudotuberculosis proteins are promising vaccine targets that need to be further evaluated, especially if a mixture of these proteins can stimulate different immune responses, a hypothesis that should be further explored in future studies. Therefore, the use of these proteins in vaccine formulations should not be revoked”, emphasizing exactly the points mentioned in the reviewers’ comment/suggestion.

Furthermore, in topic 5 (Conclusions) we added a sentence that reinforces the importance of analyzing if the proteins can trigger different immune responses. The sentence on page 9, lines 337-340 now reads: “Evaluating the cellular immune response triggered by these formulations is the next step in elucidating whether the association of recombinant proteins NanH and PknG is the right path in vaccine development for CLA, especially if the proteins manage to elicit different immune responses”.

We hope the reviewer approves of these changes. If not, please let us know and we will adjust them.

7. Minor typo should be corrected; like C. Pseudotuberculosis should be italic at lines 45, 47, and 52.

Answer: The authors would like to thank you for your careful reading of the text. We have corrected the typos mentioned above and reviewed the whole text to detect any other adjustments that need to be made.